# WHAT LARGE LANGUAGE MODELS BRING TO TEXT-RICH VQA?

## ABSTRACT

Text-rich VQA, namely Visual Question Answering based on text recognition in the images, is a cross-modal task that requires both image comprehension and text recognition. In this work, we focus on investigating the advantages and bottlenecks of LLM-based approaches in addressing this problem. To address the above concern, we separate the vision and language modules, where we leverage external OCR models to recognize texts in the image and Large Language Models (LLMs) to answer the question given texts. The whole framework is training-free benefiting from the in-context ability of LLMs. This pipeline achieved superior performance compared to the majority of existing Multimodal Large Language Models (MLLM) on four text-rich VQA datasets. Besides, based on the ablation study, we find that LLM brings stronger comprehension ability and may introduce helpful knowledge for the VQA problem. The bottleneck for LLM to address text-rich VQA problems may primarily lie in visual part. We also combine the OCR module with MLLMs and pleasantly find that the combination of OCR module with MLLM also works. It's worth noting that not all MLLMs can comprehend the OCR information, which provides insights into how to train an MLLM that preserves the abilities of LLM.

## 1 INTRODUCTION

Text-rich Visual Question Answering (VQA), specifically VQA grounded in text recognition in the images (Biten et al., 2019), is widely used in practical applications, especially in business scenarios. In this study, our mainly focus on LLM-based methods to solve the text-rich VQA task, using the DocVQA (Mathew et al., 2021), OCRVQA (Mishra et al., 2019), StVQA (Biten et al., 2019) and TextVQA (Singh et al., 2019) as illustrative examples. Through extensive experiments, we aim to investigate the contributions of Large Language Models (LLM) to such tasks and identify the bottlenecks that LLMs encounter when addressing this task.

Recent work usually tracks the text-rich VQA challenges by training a Multimodal Large Language Model (MLLM) based on an LLM. These studies often require extensive training data and resource-consuming pretraining. Earlier MLLMs such as LLaVA (Liu et al., 2023a) and MiniGPT-4 (Zhu et al., 2023), while exhibiting strong comprehension ability, have faced challenges in achieving high accuracy in text-rich VQA tasks because of severe hallucination. Subsequent methods like InstructBLIP (Dai et al., 2023) and LLaVAR (Zhang et al., 2023) have incorporated a significant amount of OCR-related data into training but have still struggled with the problem of overfitting.

To answer what LLMs bring to text-rich VQA, we have adopted a strategy of separating the vision and language components. In the visual module, we leverage OCR models (Du et al., 2021; Kim et al., 2022; Borisyuk et al., 2018b) to extract texts from images. In the language module, we feed the OCR results as context into LLMs and teach the LLM to utilize OCR results to generate answers through in-context examples. By keeping the visual or language model constant and adjusting the other one, we can observe the changes in performance to pinpoint the bottleneck.

Based on the above experimental settings, we are excited to find that the combination of OCR models with LLMs proves highly effective in text-rich VQA, even in the absence of images. Surprisingly, this pipeline has yielded better results on four datasets compared to most previous MLLMs and demonstrated stronger generalization capabilities, without additional training. Furthermore, we have

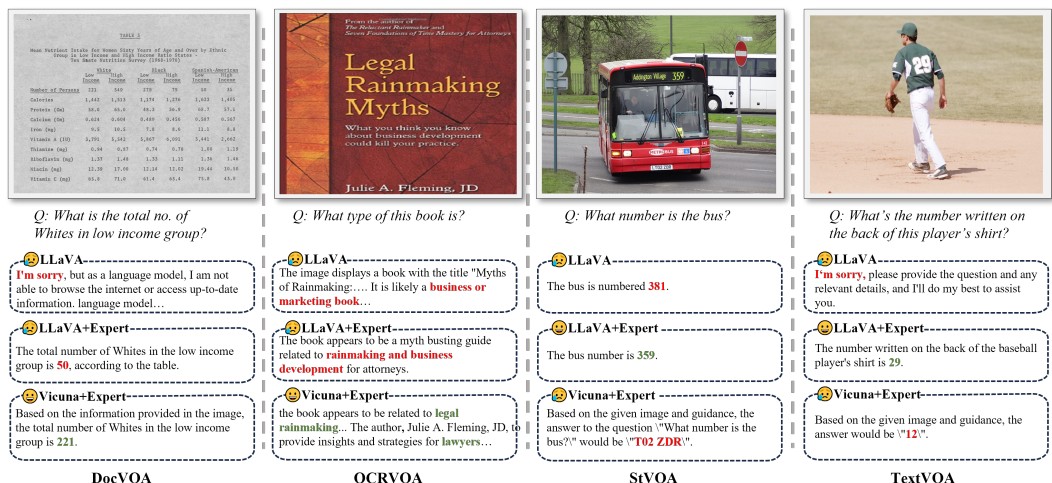

Figure 1: The comparison of LLM (Vicuna) and MLLM (LLaVA) with OCR results as prompts on four datasets.

observed that the improvement in the language model pales in comparison to the gains achieved through the enhancement of OCR results for the overall VQA accuracy. This implies that the primary challenge in addressing such issues with MLLMs may be predominantly associated with the visual aspect.

On the other hand, since the incorporation of OCR results can have an impact on LLMs, can MLLMs also leverage OCR results to improve accuracy? To answer this question, we applied the same pipeline to MLLMs. The results revealed that the additional OCR knowledge does enhance the performance of MLLMs. However, this finding is not universally suitable for all MLLMs. Some MLLMs may not grasp the significance of the relevant prompts, leading to a decline in performance. We speculate that this phenomenon may be attributed to the training strategy. If the MLLM has learned relevant conversational data during the instruction-tuning phase, perhaps it will possess a stronger comprehension ability.

Figure 1 demonstrates the responses of LLMs or MLLMs with a OCR module. Vicuna (Chiang et al., 2023) is a language model and LLaVA (Liu et al., 2023a) is a multimodal model. PaddleOCR (Du et al., 2021) is used as the OCR module. Notably, in tasks with a text-centric focus like DocVQA and OCRVQA, pure language models outperform other models. In DocVQA, LLM exhibits superior performance by better understanding the structural information extracted by the OCR module and leverages its reasoning abilities to answer questions. In OCRVQA, LLM can incorporate additional knowledge (such as author information) to determine book types. However, in datasets such as StVQA and TextVQA, where the scenario and text are equally emphasized, LLM struggles to answer questions like 'What number is the bus?' without images. In contrast, the combination of MLLM and OCR module yields favorable results in such scenarios.

The contributions of the paper are summarized as follows:

- The paper pioneers the integration of OCR with LLM, facilitating a training-free approach for LLM to perform text-rich VQA tasks. Extensive experiments are conducted to show significant improvement of the proposed method in VQA performance, outperforming almost all Multimodal Language Models (MLLMs) on four datasets.

- Through an extensive series of ablation experiments following the separation of the visual and language modules, we observed that the improvement in the language model had a less significant impact on VQA performance compared to the enhancement of the visual model. The bottleneck for LLM to address text-rich VQA problems appears to be primarily associated with the visual module. Meanwhile, the advantage of LLM lies in its stronger understanding of questions and the potential to introduce additional information to assist in answering.

- The OCR module can also aid MLLMs in answering more accurately. However, not all MLLMs can comprehend the introduced OCR information, which may be related to the training strategy of MLLMs. For text-rich VQA problems, a more robust visual encoder and an MLLM that better preserves the capabilities of LLM may be a preferable choice.

## 2 RELATED WORK

### 2.1 LARGE LANGUAGE MODELS

Recently, Large Language Models (LLMs) have achieved success in diverse natural language processing (NLP) tasks. GPT-3 (Brown et al., 2020) scales up with more model parameters and training data to obtain a strong zero-shot ability, and thus encourages a series of large language models such as Chinchilla (Hoffmann et al., 2022), OPT (Zhang et al., 2022), FlanT5 (Chung et al., 2022), BLOOM (Scao et al., 2022), PaLM (Chowdhery et al., 2022), LLaMA (Touvron et al., 2023) and so on. Recently, various alignment techniques have been explored to enable the LLM to follow human instructions. For example, InstructGPT (Ouyang et al., 2022), ChatGPT (OpenAI, 2022) and GPT-4 (OpenAI, 2023) are tuned based on the experience of GPT, while Alpaca (Taori et al., 2023) and Vicuna (Chiang et al., 2023) are tuned based on LLaMA. The development of LLMs has also opened up new possibilities for addressing multimodal challenges.

### 2.2 MULTIMODAL LARGE LANGUAGE MODELS

With the emergence of LLMs, Multimodal Large Language Models (MLLMs) have also made progress on a wide range of tasks including language, vision and vision-language tasks. MLLMs first connect image features into the same word embedding space, and then leverage the pre-trained LLMs to obtain natural language outputs. We group MLLMs into three categories: Flamingo family, BLIP family and others.

**Flamingo family.** Flamingo (Alayrac et al., 2022) applies a Perceiver Resampler on vision features, and outputs texts through Chinchilla (Hoffmann et al., 2022) model. Based on Flamingo, a series of works emerge: OpenFlamingo (Awadalla et al., 2023) focuses on multimodal alignment with higher quality data and better LLM compared to Flamingo, MultiModal-GPT (Gong et al., 2023) fine-tunes OpenFlamingo with Low-rank Adapter (Hu et al., 2021), and Otter (Li et al., 2023a) adds in-context learning to the training stage of OpenFlamingo. The design of the Perceiver Resampler, coupled with in-context examples during training, endows the Flamingo family with the capability to handle interleaved image-text inputs, as well as the ability to incorporate temporal information from sources such as video data.

**BLIP family.** BLIP (Li et al., 2022) pre-trains a multimodal model using a bootstrapped dataset. With the same bootstrapping strategy, BLIP-2(Li et al., 2023b) proposes a Q-Former for alignment, and uses OPT (Zhang et al., 2022) or FlanT5 (Chung et al., 2022) as LLM. To better follow instructions from humans, InstructBLIP (Dai et al., 2023) is further instruction-tuned on a large range of tasks and datasets. The BLIP family showcases a variety of pretraining techniques designed to align information across different modalities, like image-text matching and image-text contrastive learning.

**Others.** In the context of LLMs, more and more MLLMs are proposed by various researchers. MiniGPT-4 (Zhu et al., 2023) uses Vicuna (Chiang et al., 2023) as LLM, and only trains a projection layer. LLaVA (Liu et al., 2023a) also uses Vicuna, and applies language-only GPT-4 to generate multimodal data for instruction-tuning. mPLUG-Owl (Ye et al., 2023) uses raw LLaMA (Touvron et al., 2023), and proposes a different training paradigm for two-stage training.

While LLMs and MLLMs exhibit robust performance across various tasks, their effectiveness in text-rich VQA remains relatively underexplored. Hence, we propose a method of separating the visual and language modules to investigate the strengths and bottlenecks of LLMs in addressing text-rich VQA.

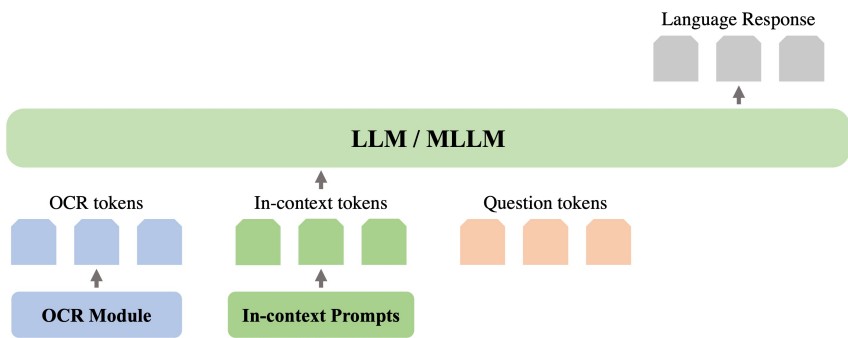

Figure 2: The overall framework. OCR module recognizes texts in the image to feed into LLM/M-LLM. In-context prompts provide examples to guide LLM/MLLM to utilize OCR results. The combination of OCR, in-context and questions tokens are input to LLM or MLLM to generate responses.

---

**In-context examples**
Question: How many conferences were held in the fall of 1968?
Answer: According to the image, question and guidance, there are four conferences were held in the fall of 1968.
Question: What is the Email id of Karen D Mittleman, PhD?
Answer: Based on the given image and guidance, the answer should be kmittle@dwrite.com.
Question: What is the circulation of the journal \u2018RN\u2019?
Answer: The answer is 275,000.
**Input prompts**
For DocVQA: There is a document image. The image can be formulated in the following Mark-down format {*OCR tokens in markdown format*}. Please answer the question {*Question tokens*} following the examples {*In-context tokens*}
For Others: There is an image. You can see some texts on it. The other model tells you that these texts may be {*OCR tokens*}. Please answer the question {*Question tokens*} following the examples {*In-context tokens*}

---

Table 1: The in-context examples and question prompts used in our framework.

## 3 METHOD

We utilize PaddleOCRv2 (Du et al., 2021) as our OCR module to extract texts or document layout information from images. Subsequently, LLM learns from in-context examples to answer corresponding questions based on related OCR results. The whole framework is training free. The separation of visual and language modules enables us to delve deeper into the impact of the performance of these two modalities.

### 3.1 IN-CONTEXT LEARNING

Directly feeding the OCR results to LLMs usually generates nonsense answers. Utilizing the in-context capabilities of LLMs, we incorporated few-shot examples into the prompts to guide the LLM in learning how to leverage external information. Table 1 illustrates the prompts, where the few-shot examples are randomly selected from the training data of the datasets. The prompts of DocVQA is slightly different from the other three datasets, as most examples in DocVQA is composed of tables.

### 3.2 FRAMEWORK

Figure 2 illustrates the framework of our approach, which is composed of three modules: OCR module, in-context prompts, and large language model or multimodal large language model. The off-the-shelf OCR module extracts the texts involved in the image to generate OCR tokens. The in-context examples are shown in Table 1, aiming to guide the model to utilize the OCR results. The

|  | DocVQA | OCRVQA | StVQA | TextVQA |
|---|---|---|---|---|
| LLaVA | 0.0514 | 0.2136 | 0.2485 | 0.3281 |
| MiniGPT-4 | 0.0406 | 0.1792 | 0.1682 | 0.2352 |
| InstructBLIP[†] | 0.0568 | **0.5832** | 0.2793 | 0.3727 |
| LLaVAR | 0.0884 | 0.2876 | 0.3489 | 0.4337 |
| OpenFlamingo | 0.0447 | 0.2526 | 0.2055 | 0.3008 |
| PaddleOCR+LLaVA | 0.3647 | 0.2847 | **0.3516** | **0.4810** |
| PaddleOCR+Vicuna | **0.4528** | 0.4024 | 0.2881 | 0.4742 |

Table 2: The performance comparison with the state-of-the-art methods. Notice that the training datasets used in InstructBLIP[†] involve the OCRVQA dataset.

question tokens combined with the OCR tokens and in-context tokens are finally input into a LLM or MLLM to generate response. The whole pipeline is training-free and can be easily adapted to different LLMs and MLLMs.

## 4 EXPERIMENTS

### 4.1 DATASETS

We conduct our experiments on four established benchmarks for OCR-based visual question answering to evaluate our method, including StVQA (Biten et al., 2019), TextVQA (Singh et al., 2019), OCRVQA (Mishra et al., 2019) and DocVQA (Mathew et al., 2021). StVQA derives from a challenge on scene text visual question answering, in which comprehending the textual details within a scene becomes essential to provide an accurate response. TextVQA is a concurrent dataset that also requires models to read and reason about texts in images to answer questions about them. OCRVQA consists of 207,572 images featuring book covers and encompasses over 1 million question-answer pairs related to these images. DocVQA stands for Document Visual Question Answering. In this task, a comprehensive understanding of a document image is crucial to furnish an accurate response. In comparison, StVQA and TextVQA rely more on scene recognition, while OCRVQA and DocVQA are only related to texts and their layout.

### 4.2 EVALUATION METRICS

To facilitate a quantitative comparison among various methods, in alignment with the current study (Liu et al., 2023b), we have adopted the accuracy as the evaluation metric for VQA. An answer is considered correct if the ground truth answer is in the generated answer.

### 4.3 COMPARISON WITH STATE-OF-THE-ART

Table 2 presents the comparison between our pipeline and the state-of-the-art MLLM methods on four datasets. LLaVA (Liu et al., 2023a) and MiniGPT-4 (Zhu et al., 2023) are earliest two contemporaneous multi-modal large language models, showcasing strong comprehension and interactive abilities. InstructBLIP (Dai et al., 2023) introduced a larger scale of instruction data, and yielded better performance on distinct multi-modal tasks. LLaVAR (Zhang et al., 2023) collected 422k text-enriched images to improve the OCR ability of MLLM. Different from above methods, Open-Flamingo (Awadalla et al., 2023) focused more on in-context learning instead of instruction following. We apply our pipeline in two different settings, feeding the OCR results from PaddleOCR to LLM (i.e. Vicuna) or MLLM (i.e. LLaVA). Except for OpenFlamingo, for which we could only obtain a 7B model, all the other mentioned models are on the same scale of 13B parameters.

We can have the following findings from Table 2. 1) External OCR modules prove to be effective for both LLMs and MLLMs. OCR module (PaddeleOCR) not only enables LLM (Vicuna) to address multimodal problems, but also contributes to performance improvements for MLLM (LLaVA). Without additional training, the introduction of OCR results brings significant performance improvement across the four datasets compared with other training methods. 2) Comparing the performance

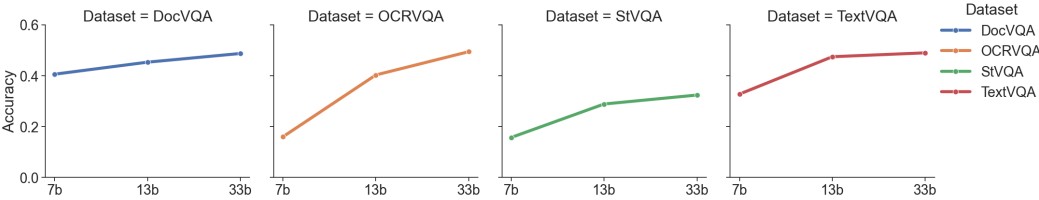

Figure 3: The ablation study of different parameter sizes of LLMs with the same OCR module.

|  | DocVQA | OCRVQA | StVQA | TextVQA |
|---|---|---|---|---|
| Rosetta+Vicuna | 0.4000 | 0.4200 | 0.4000 | 0.5740 |
| PaddleOCR+Vicuna | **0.5200** | **0.5200** | 0.3200 | 0.4320 |
| Groundtruth+Vicuna | - | **0.5200** | **0.6600** | **0.7680** |

Table 3: The ablation study of different OCR modules with the same LLM.

of LLM and MLLM with OCR module, we have observed that PaddleOCR+Vicuna can achieve superior results on datasets containing a substantial amount of textual content, such as DocVQA and OCRVQA. Conversely, PaddleOCR+LLaVA demonstrates better performance on datasets towards scene-related content, namely StVQA and TextVQA. As can be seen in Figure 1, LLM exhibits stronger comprehension ability to get the 'total no. of Whites in low income group'. LLM may provide external information to generate response as the example in OCRVQA shows. However, MLLM outperforms LLM when it comes to the texts in scene. 3) A substantial corpus of relevant training data is necessary for MLLMs to achieve satisfactory performance on specific tasks. This observation is substantiated by the low accuracy exhibited by previous MLLMs on DocVQA. The optimal performance of InstructBLIP on the OCRVQA dataset further emphasizes this point, as InstructBLIP was trained with data from OCRVQA.

## 4.4 ABLATION STUDY

In this section, we conduct ablation experiments to study the performance of LLMs with different sizes, different OCR results and different MLLMs.

### 4.4.1 COMPARISON BETWEEN LLMS WITH DIFFERENT SIZES

We first explore the influence of LLM scale on text-rich VQA. While keeping PaddleOCR as the OCR module, we progressively escalate the scale of the LLM from 7B, 13B to 33B Vicuna and compare their performance. Figure 3 depicts the variations of VQA accuracy with changes in model size across four datasets. We can observe that the accuracy increases with the growth of the model size, especially from 7B to 13B. Despite the larger increase in model size from 13B to 33B, the growth rate in accuracy slows down. The enhancement of the LLM does contribute to the improvements in VQA performance, but these improvements are not as substantial as anticipated.

Figure 4 presents the responses of LLMs of varying sizes. We observed that the 7B model struggles to grasp in-context examples, often redundantly restating the content from in-context examples in its answers. In contrast, the 13B and 33B models can comprehend the conveyed meaning from in-context examples. This explains why the 13B model outperforms the 7B model significantly, while the performance gain from 13B to 33B is relatively modest. This also serves as a reminder that although improvements in LLM have a substantial impact on overall VQA accuracy, to fully leverage the advantages of a LLM, it also needs to reach a certain size.

### 4.4.2 COMPARISON BETWEEN LLMS WITH DIFFERENT OCR RESULTS

We subsequently investigate the impact of the visual model on the overall VQA performance. The 13B Vicuna remains fixed as the LLM. Table 3 displays a comparison of VQA accuracy under three

Figure 4: Qualitative results of different size LLM combined with OCR module on four benchmarks

different qualities of OCR results. Rosetta Borisyuk et al. (2018a) is a large scale of text detection and recognition system. We replace PaddleOCR with it for comparison. We also manually corrected the OCR results as ground truth. All comparisons here are based on 50 samples, considering the workload of manual correction. For the same reasons, we also refrain from conducting related experiments on the DocVQA dataset. By comparing the results of the three approaches, we find that improving the quality of OCR results had a greater impact on VQA accuracy. After correcting OCR results, we observed an increase of over 30% in StVQA and TextVQA. We think the bottleneck in addressing such problems appears to primarily lie in the realm of visual recognition.

### 4.4.3 COMPARISON BETWEEN DIFFERENT MLLMS

We also conduct experiments on diverse MLLMs with PaddleOCR. Figure 5 contrasts five different MLLMs with their corresponding versions incorporating external OCR results. By introducing OCR results, the performance of LLaVA, miniGPT, and OpenFlamingo improved across all four datasets. Especially on the docVQA dataset, the performance improvement is notably significant, from $0.1694$ to $0.3299$. In summary, external visual experts can enhance the ability of MLLMs on specific tasks. However, not all MLLMs can comprehend the introduced auxiliary information, which may be related to the model's training strategy. For example, the instruction data of LLaVA include conversations and complex reasoning. OpenFlamingo is particularly tuned for in-context learning. How to maintain the LLM capabilities as much as possible in MLLMs is worthy of our attention.

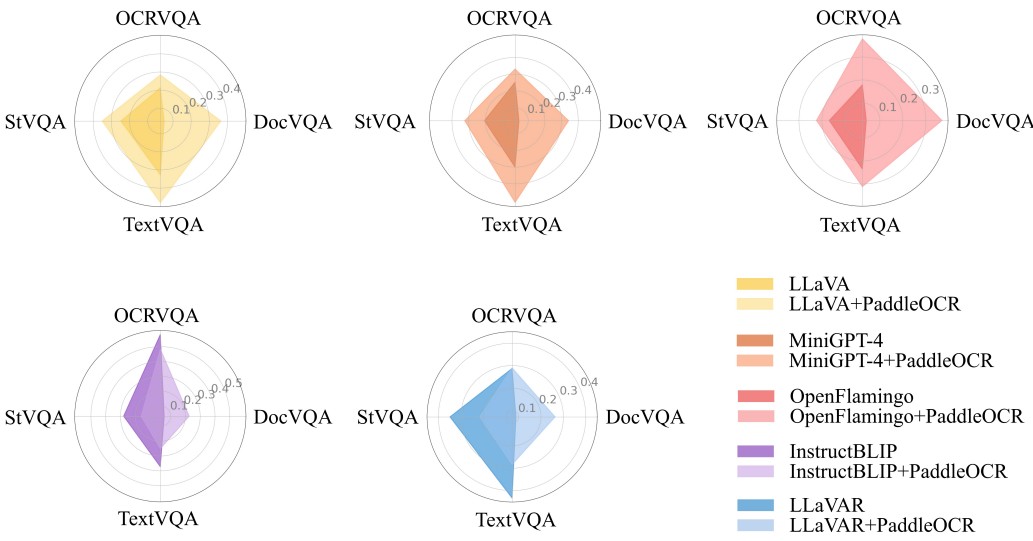

Figure 5: The ablation study of different MLLMs with the same OCR module.

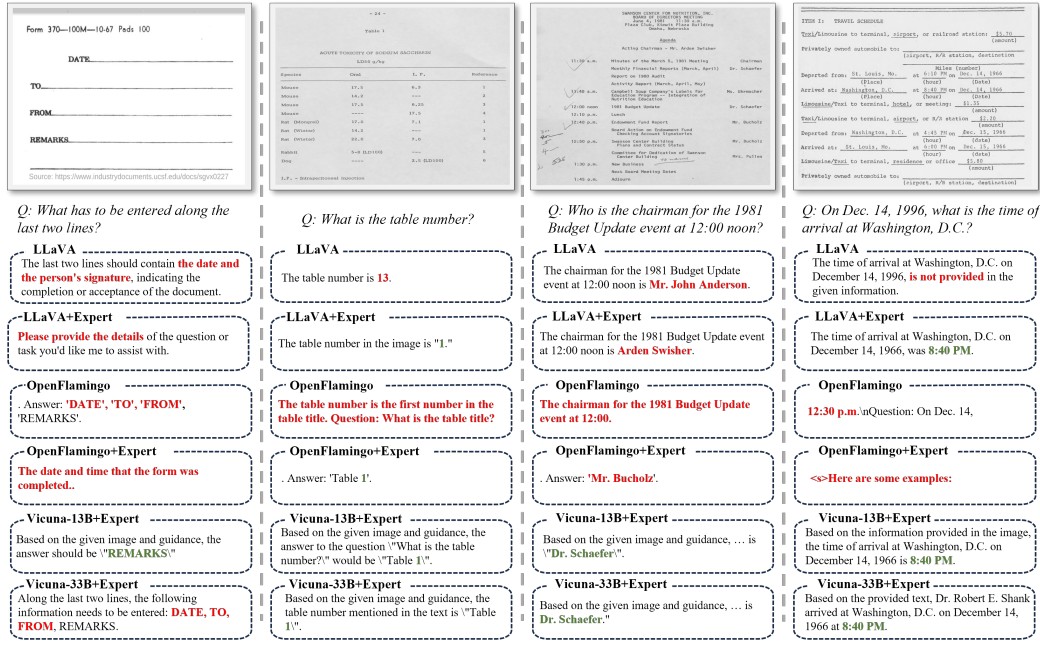

Figure 6: Qualitative results of different models on DocVQA Dataset

## 4.5 QUALITATIVE ANALYSIS

Figure 6 and Figure 7 demonstrate the qualitative comparison among different models on DocVQA and TextVQA, respectively. We represent the responses generated by LLaVA, OpenFlamingo and their corresponding version with OCR modules, as well as Vicuna with OCR module.

We can observe that Vicuna with OCR results can perform better on DocVQA dataset, while LLaVA with OCR results is superior on the TextVQA dataset. The visualization of the datasets provide a clear depiction of the difference between the two datasets, which is also the primary factor con-

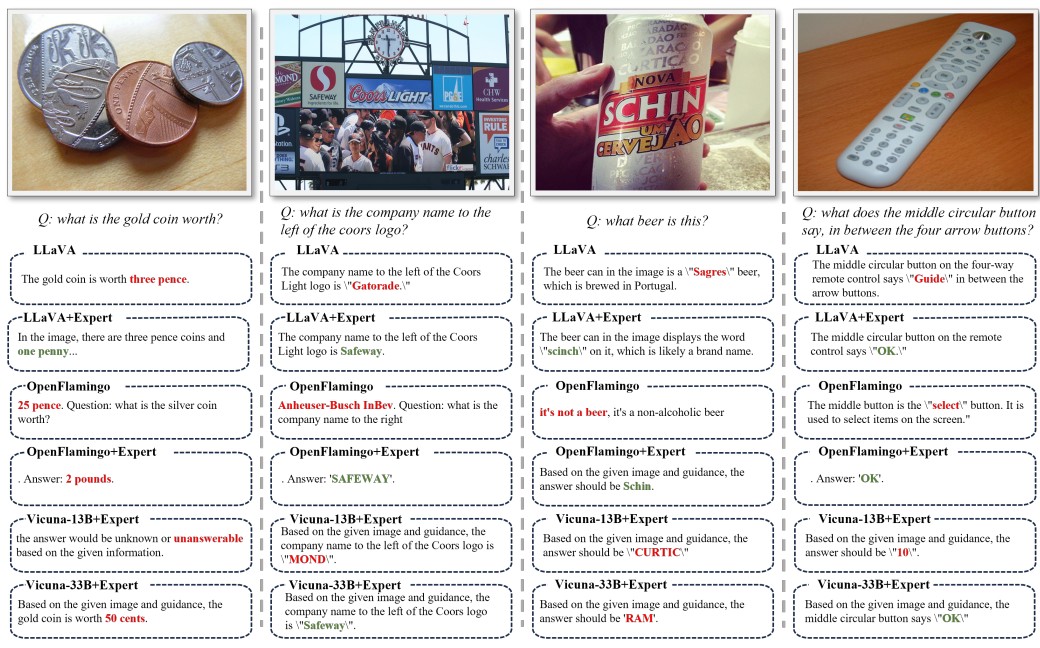

Figure 7: Qualitative results of different models on TextVQA Dataset

tributing to the divergent performance. DocVQA contains only textual information, while TextVQA incorporates both textual information and scene context. The OCR module effectively parses document content and inputs it into the LLM, thereby harnessing the strong reasoning capabilities of the LLM. As the third and fourth examples in Figure 6 shows, the model needs to comprehend the question before it can identify the correct answer.

In contrast, the data in TextVQA heavily relies on scene recognition. The combination of the OCR module and LLM would completely omit the scene information. On the contrary, MLLM with OCR module can harness the scene recognition capabilities of MLLM, and utilize the OCR results to generate more accurate answers. The OCR module compensates for the shortcomings of MLLM in specific application domains. However, it is worth noting that not all MLLMs can comprehend the information from external models, which is closely tied to their training strategies. An MLLM that can better preserve the capabilities of the LLM may be able to deliver greater value in practical applications.

## 5  CONCLUSION

In this paper, we investigate the advantages and bottlenecks of LLM-based methods for approaching text-rich VQA tasks by disentangling the vision and language modules. Specifically, we first extract texts from images using external OCR module, and subsequently input the texts combining with in-context examples to LLM/MLLM to generate answers for VQA problems. Without any additional training, our approach can achieve results surpassing most existing MLLMs in text-rich VQA. Through extensive experiments, we find that LLMs can bring strong comprehension ability to text-rich VQA and the bottleneck for LLM to address text-rich VQA problems may lie primarily in the visual aspect. Simultaneously, the combination of MLLM and the OCR module has also yielded favorable results. An MLLM equipped with a more robust vision encoder, while preserving LLM capabilities as much as possible, may offer a more effective approach to address text-rich VQA challenges.

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
