# OpenReview forum: "What Large Language Models Bring to Text-oriented VQA?"
_ICLR.cc/2024/Conference — ICLR 2024 Conference Withdrawn Submission_

### Official Review · Reviewer_qy6F · 2023-10-22

**Soundness:** 2 fair
**Presentation:** 3 good
**Contribution:** 2 fair
**Rating:** 3
**Confidence:** 4

**Summary:**

This research evaluates Large Language Models' (LLMs) effectiveness in text-rich Visual Question Answering (VQA) tasks, revealing notable strengths in comprehension but significant bottlenecks in visual processing. The study's training-free approach, utilizing external OCR for text recognition and LLMs, outshines existing Multimodal Large Language Models (MLLMs) in several datasets. However, integrating OCR with MLLMs presents inconsistencies, indicating a crucial need for enhanced visual processing in LLMs and refined OCR comprehension in MLLM training. The findings spotlight the promise of LLMs in text-rich VQA while outlining critical areas for future advancement.

**Strengths:**

1. Integration of OCR with LLM, facilitating a training-free approach for LLM to perform text-rich VQA tasks.
2. The authors observed that the OCR module can also aid MLLMs in answering more accurately.

**Weaknesses:**

1. There still are some researches that have investigated incorporating OCR into a model, such as https://anandmishra22.github.io/files/mishra-OCR-VQA.pdf. The approach of this paper is common and lacks novelty
2. Why can InstructBLIP achieve a superior performance on the OCRVQA dataset but other MLLMs can’t?
3. The results of fine-tuning the MLLMs for the text-rich VQA task do not seem to be reported in the experiment, and I wonder if the performance of fine-tuned model can be further improved with OCR.
4. More advanced LLMs should be verified by this method, such as GPT-3 and GPT-4.
5. The effect of different OCR models on the proposed model should be explored.

**Questions:**

See weakness.

---

### Official Review · Reviewer_B11c · 2023-10-25

**Soundness:** 3 good
**Presentation:** 3 good
**Contribution:** 2 fair
**Rating:** 3
**Confidence:** 4

**Summary:**

To address image comprehension and
text recognition problems of LLM, the papers employs external OCR models to recognize text within images and use Large Language Models (LLMs) to answer questions based on these texts. Extensive experiments are conducted to show
significant improvement of the proposed method in VQA performance,

**Strengths:**

The paper is clearly articulated and easy to grasp, with experiments effectively showcasing the superiority of the proposed method over baseline approaches.

**Weaknesses:**

1. The scope of this paper is limited, focusing exclusively on text-rich VQA.

2. While the method is straightforward, its potential for improvement is constrained. Specifically, even when using ground-truth OCR labels, the method doesn't deliver exceptional results. This suggests that the core issue remains unresolved.

3. The approach lacks novelty in this domain. Previous studies [1,2] have already explored VQA with zero training. However, this paper does not draw comparisons with them, further diminishing its novelty.

[1] Yang, Z., Gan, Z., Wang, J., Hu, X., Lu, Y., Liu, Z., & Wang, L. (2022, June). An empirical study of gpt-3 for few-shot knowledge-based vqa. In Proceedings of the AAAI Conference on Artificial Intelligence (Vol. 36, No. 3, pp. 3081-3089).

[2] Guo, J., Li, J., Li, D., Tiong, A. M. H., Li, B., Tao, D., & Hoi, S. (2023). From Images to Textual Prompts: Zero-shot Visual Question Answering with Frozen Large Language Models. In Proceedings of the IEEE/CVF Conference on Computer Vision and Pattern Recognition (pp. 10867-10877).

**Questions:**

As shown in the last section

---

### Official Review · Reviewer_dgPH · 2023-10-30

**Soundness:** 3 good
**Presentation:** 3 good
**Contribution:** 2 fair
**Rating:** 5
**Confidence:** 4

**Summary:**

This paper proposed a framework for text-dependent VQA tasks leveraging OCR and LLM models to extract text and perform QA + reasoning. While this approach is not entirely novel, the breadth of analysis in this work is significant. The authors highlight limitations of the MLLM category on these tasks and point to potential directions for improvement in those models.

**Strengths:**

This paper contains extensive experimentation on text-oriented vqa with multiple algorithms to clarify strengths and weaknesses of the proposed framework. It also goes beyond the core thesis of the work by examining the nature of the mllm's with ocr input, an interesting direction. The observation that "not all MLLM's can comprehend the auxilliary information" from external experts suggests concrete directions in the MLLM field (finetuning with side information from external models). OCR is a clear example of a tradeoff scenario, where for accurate OCR higher resolution inference and training will be required to improve the core model which can be eschewed in favor of finetuning with side information from an accurate OCR engine.

**Weaknesses:**

The proposed framework does not seem very novel. It is an obvious approach for scenarios like the DOCVQA and if I'm not mistaken it has been used before (See the line of work on UDOP or LayoutLMvX, where qa can be done by query encoding and treats the answering like text extraction, or T5 for a more direct analogy to the framework). The authors should highlight and clarify the differences with past approaches.

Furthermore, the reported performance is far from the reported state of the art on DOCVQA. I cannot verify the same for the other tasks, but concerns about the significance of the proposed framework come up from this.

**Questions:**

What are the key novelties of the method proposed in this work? It seems like a framework that has been used before.
What is the significance of the reported performance? It seems that the proposed framework is already obsolete on some of the domains so is there any relevance to it?

---

### Official Review · Reviewer_yJVr · 2023-11-01

**Soundness:** 3 good
**Presentation:** 3 good
**Contribution:** 2 fair
**Rating:** 3
**Confidence:** 5

**Summary:**

Submission 1224 introduces a text-rich VQA task that requires the model to recognize text and comprehend image content. The authors aim to investigate the advantages and bottlenecks of LLM-based methods in addressing the text-rich VQA task. In their work, they propose vision and language modules. Specifically, the authors leverage OCR systems such as paddleOCR and rosetta to extract text from images. Additionally, they harness the powerful in-context ability of the language model to provide answers based on extracted text. The proposed in-context learning method for the text-rich VQA task has achieved competitive performance compared to the majority of existing Multimodal Large Language Models. Furthermore, the paper pioneers the integration of OCR with LLM, facilitating a training-free approach for LLM to perform text-rich VQA tasks.

**Strengths:**

The paper is the first to utilize the in-context learning ability of large language Model and Multimodal Large Language Models for VQA tasks. Meanwhile, the proposed method has achieved competitive performance compared to the majority of existing Multimodal Large Language Models.
The paper provides a well-structured review of relevant literature in large language models and multimodal large language models, with clear motivation for the proposed solutions.

**Weaknesses:**

In Section 3, the proposed method utilizes simple input prompts as in-context learning examples, which leads to the performance improvement primarily attributed to the in-context ability of Multimodal Large Language Models and Large Language Models. Consequently, the paper lacks substantial innovation and mainly focuses on the utilization of prompt engineering tricks for text-rich VQA.
In section 4.2, since a substantial portion of text-rich VQA answer are derived from OCR text, the evaluation metrics have significant issues where an answer is considered correct if the ground truth answer is in the generated answer. If we regard all OCR text in the image as predicted answer, the performance may also be competitive.
In section 4.3 comparison with SOTA , there seems to be an issue with the comparison between the SOTA methods. The proposed method utilizes few samples to construct in-context examples, but it is unclear whether the SOTA methods also utilize an equivalent number of samples for in-context learning or fine-tuning.
In section 4.3 comparison with SOTA, there is a suspicion of setting a low baseline when comparing with SOTA methods, for instances LLaVAR reaching 48.5 for textvqa, InstructBLIP reaching 50.7 for textvqa in original paper.

**Questions:**

In Section 3, the proposed method utilizes OCR tokens, question and answer as in-context examples. If the authors incorporate object information, such as spatial relationships between OCR within scene images, it would be a valuable addition to their work.
In section 4.2, since a substantial portion of text-rich VQA answer are derived from OCR text, the evaluation metrics have significant issues. If authors provide answer accuracy and answer ANLS as evaluation metrics, it would be appropriate to revise the manuscript score.